



# Technical note: Novel analytical solution for groundwater response to atmospheric tides

Jose M. Bastias Espejo[1], Chris Turnadge[2], Russell S. Crosbie[2], Philipp Blum[1], and Gabriel C. Rau[3]

[1]Institute of Applied Geosciences (AGW), Karlsruhe Institute of Technology, Karlsruhe (KIT), Germany
[2]CSIRO Land and Water, Adelaide, Australia
[3]School of Environmental and Life Sciences, The University of Newcastle, Callaghan, Australia

**Correspondence:** Jose M. Bastias Espejo (jose.bastias@kit.edu)

**Abstract.**

    Subsurface hydraulic and geomechanical properties can be estimated from well water level responses to Earth and atmospheric tides. However, the limited availability of analytical solutions restricts the applicability of this approach to realistic field conditions. We present a new and rigorous analytical solution for modelling flow between a subsurface-well system caused by
harmonic atmospheric loading. We integrate this into a comprehensive workflow that also estimates subsurface properties using a well-established Earth tide method. When applied to groundwater monitoring datasets obtained from two boreholes screened in a sand aquifer in the Mary-Wildman Rivers region (Northern Territory, Australia), estimated hydraulic conductivity and specific storage agree. Results also indicate that small vertical leakage occurs in the vicinity of both boreholes. Furthermore, the estimated geomechanical properties were within the values reported in literature for similar lithological settings. Our new
solution extends the capabilities of existing approaches, and our results demonstrate that analysing the groundwater response to natural tidal forces is a low-cost and readily available solution for unconsolidated, hydraulically confined, and undrained subsurface conditions. This approach can support well-established characterisation methods, increasing the amount of subsurface information.

## 1 Introduction

Knowledge of subsurface hydro-geomechanical properties is crucial for Earth resource development and management. Such properties determine the capacity of hydrostratigraphic units to store and transmit groundwater. Traditional, active hydraulic testing methods such as pumping, slug, pressure and packer tests or laboratory analyses of cores involve considerable logistical expenses (Maliva, 2016). In contrast, passive methods (e.g. *Tidal Subsurface Analysis*, TSA), which are used to estimate hydraulic properties from well water level responses to ubiquitous periodic forces (Merritt, 2004; Cutillo and Bredehoeft, 20  2011), are relatively low cost to implement and derive additional value from commonly measured datasets (McMillan et al., 2019; Rau et al., 2020, 2022). The effect of gravitational effects and atmospheric loading on the subsurface has been long observed and reported (Meinzer, 1939) and is contained in routine groundwater pressure measurements made in countless observation wells around the world (McMillan et al., 2019). The influence of natural forces such as tides on groundwater pressures are ubiquitous allowing widespread application reducing effort and cost of investigations. Since passive approaches



rely on natural signals and do not require any active perturbation of the subsurface system, we will refer to them collectively
as *Passive Subsurface Characterisation* (PSC) in our work.

Earth and atmospheric tides act at as harmonic forces at various frequencies (McMillan et al., 2019). For groundwater
investigation the most informative frequencies range from 0.8 to 2.0 cycles per day (cpd) (Merritt, 2004). Dominant frequencies
present in groundwater pressure measurements are the $S_1$ (1.0 cpd), $M_2$ (1.93 cpd) and the $S_2$ (2.00 cpd). These components
generally show a higher amplitude in comparison with other tidal harmonics and are, therefore, more likely to be contained in
field datasets (McMillan et al., 2019).

Loading forces cause mechanical deformation of the water-saturated porous medium, leading to an instantaneous pore
pressure response and a hydraulic gradient towards the nearby observation well. This gradient drives groundwater exchange
between the subsurface and the well until re-equilibrium is achieved (Cheng, 2016; Verruijt, 2013; Wang, 2017). The am-
plitude ratio between the magnitude of well water level variation and subsurface pore pressure variation, as well as the time
delay required for groundwater exchange, expressed as a phase shift or phase lag, can be used to estimate subsurface hydro-
geomechanical properties (Hsieh et al., 1987). Positive phase shifts (i.e., when well water levels respond before subsurface wa-
ter pressures to Earth tide-induced strain variations) have been linked to vertical connectivity with adjoining hydrostratigraphic
units (Roeloffs et al., 1989). Amplitude ratios and phase shifts can be readily extracted from measurements and inverted using
established analytical solutions (McMillan et al., 2019).

Cooper Jr et al. (1965) derived an analytical solution for the movement of groundwater caused by seismic waves in fully
confined aquifers. Bredehoeft (1967) proposed a method to interpret the effect of Earth tides on observation wells based on
classic solid mechanics, which allowed the estimation of specific storage of the aquifer if the Poisson's ratio of the porous
medium was known. However, this method did not comply with Biot consolidation theory (Biot, 1941) as it did not couple
fluid dynamics with mechanical deformation. Subsequently, many studies described the effect of Earth tides in poroelastic
systems (Bodvarsson, 1970; Robinson and Bell, 1971; Arditty et al., 1978; Van der Kamp and Gale, 1983), but did not consider
the damping effect of the observation well on the amplitude and phase. To address the signal diminishing effect of a well,
Hsieh et al. (1987) combined the poroelastic response of a confined aquifer with Cooper Jr et al. (1965)'s work and derived an
analytical solution to model flow to wells due to Earth tides.

Rojstaczer (1988) proposed an analytical solution for modelling flow to wells induced by atmospheric tides. However, the
solution requires knowledge of vadose properties which are often unknown, and it does not account for the effect of barometric
efficiency on confined pore pressure (also known as tidal efficiency). To address this, Rojstaczer and Riley (1990) developed
an analytical solution that includes the barometric effects on confined pore pressure, but it does not consider the effects on
amplitude and phase shift of a well. Additionally, the mean stress in their formulation only considers the vertical direction and
neglects lateral directions, which can lead to significant errors for typical Poisson's ratio values (Cheng, 2016; Verruijt, 2013;
Wang, 2017).

Several studies have estimated subsurface properties using Earth tide analysis (Le Borgne et al., 2004; Doan et al., 2006;
Cutillo and Bredehoeft, 2011; Lai et al., 2013, 2014; Rahi and Halihan, 2013; Xue et al., 2016; Shi and Wang, 2016; Acworth
et al., 2016). However, many of the analytical solutions used to derive estimates assume oversimplified settings, which can





lead to inaccurate results. To address this, Wang et al. (2018) recently developed an analytical solution that describes flow in and out of a well caused by Earth tides in a two-layered flow system. Gao et al. (2020) accounted for the well skin effect, which occurs when the physical properties of the formation in a larger area around a borehole are affected by drilling, leading to reduced amplitude ratio and phase shift. Additionally, Guo et al. (2021) derived an analytical solution to describe flow in fractures caused by Earth tides and estimated hydraulic properties. Finally, Liang et al. (2022) solved Richards equation (Freeze

and Cherry, 1979) to include the effect of the unsaturated zone, finding that it delays the phase shift response of the borehole pressure.

     Xue et al. (2016) and Rau et al. (2020) used the analytical solution of Wang (2017) to model the barometric effect of atmospheric tides with vertical leakage, but it lacks the damping effect of an observation well. Recently, Rau et al. (2022) proposed a new approach based on the work of Acworth et al. (2017) that combines poroelastic relations for one-dimensional

Earth and atmospheric tide deformation to obtain a system of equations with poroelastic properties. However, their approach is based on an analytical model that does not correctly represent vertical leakage. To the best of our knowledge, there is no rigorous analytical solution in the literature to model flow to wells induced harmonically from atmospheric tides based on the mean stress flow equation while considering a semi-confined aquifer.

     The objective of this work is twofold. Firstly, we introduce a new analytical solution based on the Biot theory of consolidation

that describes the flow between a subsurface-well system caused by the harmonic loading of atmospheric tides. Secondly, we demonstrate its usefulness by applying it to well water levels from two boreholes in the Northern Territory of Australia and comparing the results with established Earth tide methods and existing knowledge of the groundwater system. Our study demonstrates that our new analytical solution extends the range of properties that can be accurately estimated and provides a better understanding of subsurface processes and properties.

## 80   2   Analytical solution

In this section, a new analytical solution based on the mean stress flow equation is derived to simulate flow to wells resulting from atmospheric tides loading the surface. The fluid continuity equation in the mean stress form can be used to describe the water flow from a semi confined aquifer towards an observation well. If only radial flow is assumed and small vertical fluid exchange from the semi confined layer occurs, the equation reads (Cheng, 2016; Verruijt, 2013; Wang, 2017)

$$S_\sigma H_a \left( \frac{\partial h}{\partial t} - \frac{\alpha}{3KS_\sigma} \frac{\partial \sigma}{\partial t} \right) = T \left[ \frac{\partial^2 h}{\partial r^2} + \frac{1}{r} \frac{\partial h}{\partial r} \right] - \frac{k_l}{H_l} h. \tag{1}$$

Here, $\sigma$ is the mean stress; $K$ is the drained bulk modulus of the solid material; $r$ radius; hydraulic head of the fluid (groundwater for this study), $h$ is being used as a proxy for pore pressure $p_f = \rho g h$; $T$ is the transmissivity of the aquifer with $T = k_a H_a$, where $k_a$ is the hydraulic conductivity and $H_a$ the aquifer thickness. If the aquifer is overlain by a leaky aquitard, then the downward leakage flux can be described as $k_l h H_l^{-1}$, where $k_l$ is the vertical hydraulic conductivity and $H_l$ is the aquitard

saturated thickness. Note that this approximation is only valid when $k_a k_l^{-1} \gg 1$. $\alpha$ is the Biot coefficient which is equal to one for unconsolidated systems (e.g., gravels, sands and clays), and ranges between $n \leq \alpha \leq 1$ for consolidated systems (e.g.



bedrock); where $n$ is effective porosity. $S_\sigma$ is the Biot modulus at constant stress (also known as three dimensional storage coefficient) (Cheng, 2016; Verruijt, 2013; Wang, 2017)

$$S_\sigma = \frac{\rho g}{R}, \tag{2}$$

where $g$ is gravitational acceleration and $\rho$ the fluid density. $R$ is the Biot modulus at constant stress defined as (Cheng, 2016; Verruijt, 2013; Wang, 2017)

$$\frac{1}{R} = \frac{n}{K_f} + \frac{\alpha - n(1-\alpha)}{K}, \tag{3}$$

where $K_f$ is the bulk modulus of the fluid ($K_f = 2.2 \cdot 10^9 \ Pa$ for freshwater).

Barometric pressure fluctuations cause loading at the ground surface which results in vertical deformation of the subsurface

and, therefore, changes to the internal stress balance of the fluid-solid skeleton system. For example, when atmospheric pressure rises and, the formation undergoes compressive stress resulting in an increased in the confined pore pressure (Domenico and Schwartz, 1997).

In a fully saturated porous medium, this effect can be described by Biot consolidation theory as follows (Cheng, 2016; Verruijt, 2013; Wang, 2017)

$$p_f = R\left(-\frac{\alpha}{K}\sigma + \xi\right). \tag{4}$$

Here, $p_f$ is the fluid (i.e., water in this study) pore pressure; $\sigma$ is the mean stress; $K$ is the drained bulk modulus of the solid material; $\xi$ is the change in fluid content, can be used to quantify changes in pore pressure resulting from hydraulic gradients (Cheng, 2016; Verruijt, 2013; Wang, 2017). The sign of this parameter indicates whether a fluid is leaving or entering a given porous medium.

Biot's consolidation theory assumes $\xi = 0$ when undrained conditions apply within the porous medium. Conversely, system conditions are drained when $\xi \neq 0$. Note here that a drained porous medium conceptually differs from a confined aquifer and these concepts that are often mixed up in the literature. For example, a confined aquifer may exchange fluid via one of its horizontal boundaries such as a confined aquifer bounded by a river.

We solved Eq. 1 for steady state conditions to obtain the periodic water level in an open borehole $h_w^{AT} = h_{w,o}^{AT} e^{i\omega t}$ due to

atmospheric loading, where $\omega$ is the angular frequency of the tide signal and superscript $AT$ stands for atmospheric tides, for example $S_1$ at 1 cycle per day (CPD) or the atmospheric response to $S_2$ at 2 cpd (Merritt, 2004; McMillan et al., 2019).

As boundary condition, the hydraulic head far away from the radius of influence of the borehole is given only by the mechanical response of the system

$$t > 0, r = r_\infty : h(r,t) = h_\infty = \frac{p_{f,\infty}}{\rho g}, \tag{5}$$

where $r_\infty$ is a distance far away from the radius of influence of the borehole and the hydraulic head at the borehole screen, $h_w$, should be the water level in the bore

$$t > 0, r = r_w : h(r,t) = h_w^{AT}(t), \tag{6}$$





the bore and the aquifer are free to exchange groundwater, i.e.,

$$t > 0, r = r_w : 2\pi r_w T(\partial h / \partial r) = \pi r_c^2 (\partial h_w^{AT} / \partial t). \tag{7}$$

With these boundary conditions the solution of the water level in the borehole is derived as

$$h_{w,o}^{AT} = \frac{i\omega H_a}{(i\omega(S_\epsilon + \frac{\rho g}{K})H_a + k_l/H_l)\gamma_a} \left(\frac{\sigma}{3K}\right), \tag{8}$$

where the periodic atmospheric loading is assumed only vertical (i.e. $\sigma = \sigma_{zz}$) modelled as $\sigma = \sigma_{atm}e^{i\omega t}$, thus $\sigma_{atm}$ represents the amplitude of the atmospheric tide, and

$$\gamma = 1 + \left(\frac{r_c}{r_w}\right)^2 \frac{i\omega r_w}{2T\beta} \frac{K_0(\beta r_w)}{K_1(\beta r_w)}. \tag{9}$$

Here, $K_0$ and $K_1$ are the modified Bessel functions of the second kind and order zero and one, respectively, and

$$\beta = \left(\frac{k_l}{TH_l} + \frac{i\omega(S_\epsilon + \frac{\rho g}{K})H_a}{T}\right)^{0.5}. \tag{10}$$

Note that $S_\epsilon$, the specific storage at constant strain, and $S_\sigma$ are related as (Cheng, 2016; Verruijt, 2013; Wang, 2017)

$$S_\epsilon = S_\sigma - \frac{\rho g}{K}. \tag{11}$$

Since it is assumed that the borehole is open to the atmosphere, any change in barometric pressure will also play a role in

the hydrostatic pressure inside the borehole. Thus, the amplitude ratio between the atmospheric loading and the confined pore pressure due to atmospheric tides, $A^{AT}$, has to be expressed as the balance between the far field pore pressure ($p_{f,\infty}(\rho g)^{-1}$, Eq. 4), the amplitude of the atmospheric load ($\sigma_{atm}(\rho g)^{-1}$) and the change of fluid level in the well, $h_{w,o}$, such as

$$A^{AT} = \left| \frac{p_{f,\infty} - \sigma_{atm} - (\rho g)h_{w,o}^{AT}}{\sigma_{atm}} \right|. \tag{12}$$

The time lag between the far field confined pore pressure and the actual change of fluid in an open well is given by

$$\Delta\phi^{AT} = arg\left(\frac{p_{f,\infty} - \sigma_{atm} - (\rho g)h_{w,o}^{AT}}{\sigma_{atm}}\right). \tag{13}$$

Note that the applied amplitude of the periodic stress at a boundary has to be equal to the atmospheric pressure

$$\sigma_{atm} = -P_{atm}, \tag{14}$$

where $P_{atm}$ is the barometric pressure measured in the field. In this convention, the compression stress is opposite in sign compared to the atmospheric pressure as an increase compresses the subsurface.

We have named our novel analytical solution as the *mean stress solution*. Drawing an analogy to the established Earth tide methods (Hsieh et al., 1987; Wang et al., 2018, e.g.,), our new solution enables the estimation of subsurface hydraulic and geomechanical properties from atmospheric tidal components that are ubiquitous in standard field measurements of well water levels. This innovative solution expands the scope of existing approaches that passively characterise subsurface processes and properties (McMillan et al., 2019).



## 3    Field application

### 3.1    Field site and groundwater monitoring

In this section, we apply our analytical solution to field data, compare the results with those derived from established Earth tide methods and consider the results in the context of existing knowledge such as from lithological logs and hydraulic testing. The overall workflow applied in this section is shown in Fig. 1 which incorporates established Earth tide methods alongside our new solution.

The study area is bounded by the Mary River National Park in the west and by the Kakadu National Park in the east. The intervening area has been of interest for irrigated agricultural development since the 1980s. The area features a sub-equatorial climate, with the dry season occurring between May and September and the wet season occurring between October and April. The highest annual mean air temperatures are recorded between October and December at around $35°C$ and the lowest in July at around $16°C$ (Tickell, 2017).

Two main hydrostratigraphic units are present as layers in the study area: (1) Mesozoic/Cenozoic sediments underlain by (2) the Proterozoic Koolpinyah Dolostone, and silt- and sandstones (Fig. 2a) (Tickell, 2017). Groundwater and mineral exploration wells are the main source of geological information as outcrops are rare. Mesozoic/Cenozoic sediments consist of unconsolidated to poorly consolidated sands, clayey sands, and clays. Lithological logs indicate that this unit is laterally extensive across the study area (Tickell, 2017). A leaky sandy clay aquitard partially confines a second semi-confined sand aquifer (B1 and B2 in Fig. 2c). This aquifer is sufficiently permeable to allow recharge to the semi-confined sand aquifer, as observed by increases in the groundwater level during each wet season. The Proterozoic strata consist primarily of Koolpinyah Dolostone and Wildman Siltstone. The hydrological behaviour of this unit is conceptually a fractured aquifer (Tickell, 2017). Constant rate discharge pumping tests indicate that the hydraulic conductivity of the Mesozoic/Cenozoic strata ranges from $8.0 \cdot 10^{-5}$ to $6.3 \cdot 10^{-4}$ $ms^{-1}$ (Appendix B).

Groundwater monitoring datasets from two boreholes B1 and B2 were analysed in this work (Fig. 2b and Table 1). Note that the original nomenclature from the Australian Northern Territory (NT) was modified (Table 1). The lithological logs indicate that the boreholes are screened in the upper strata (Fig. 2c). In general, the upper two thirds of the profile are clays and sandy clays that confines the underlying aquifer. The lower third often consists of sands, clayey sands and gravels. Sands are mostly present as fine-grained quartz with limited occurrences of coarse sands to pebbles.

Well water levels were monitored hourly between June 2016 and September 2019 in each borehole using InSitu Level TROLL 400 data loggers (InSitu Inc., USA). The measured pressure heads were converted to hydraulic head values by referencing the dips of depth to water level manually to the surveyed top of casing elevations. Concurrently, barometric pressure was recorded from September 2016 to October 2017 using an InSitu BaroTROLL 500 data logger (InSitu Inc., USA).

### 3.2    Extraction of tidal responses

Earth tide strains, barometric pressure and hydraulic heads in wells B1 and B2, are shown in Fig. 3. Outliers were identified using Pearson's rule, i.e., values that deviate more than three times the *Median Absolute Deviation* (MAD) (Pham-Gia and



**Figure 1.** Overview of the workflow applied to estimate subsurface hydraulic and geomechanical properties using the groundwater response to Earth and atmospheric tides. The data set and Python scripts developed for this work are available in an external repository (see Code and Data Availability statements).



**Table 1.** Groundwater well construction information, reference datum Geocentric Datum Of Australia (GDA) 1994. DD stands for *decimal degrees*.

| Borehole NT ID | Borehole | Latitude $[DD]$ | Longitude $[DD]$ | Total depth $[m]$ | Screen length $[m]$ | Radius $[m]$ |
|---|---|---|---|---|---|---|
| RN039769 | B1 | -12.6077 | 131.8295 | 43.0 | 4 | 0.156 |
| RN024762 | B2 | -12.6259 | 131.8801 | 61.1 | 6 | 0.203 |

Hung, 2001), and removed from the data (Fig. 1b and A1). Note that the overall head varies by about 2 m reflecting the wet and dry seasons that are typical for tropical Australia. Earth tide strains were calculated using *PyGTide* (Rau, 2018) which is
based on the widely used ETERNA PREDICT software Wenzel (1996) (Fig. 1c).

Harmonic tidal components of the ten dominant target frequencies between 0.33 and 2.2 cycles per day (cpd) (Merritt, 2004; McMillan et al., 2019) were extracted from all time series and locations following the methods outlined in Schweizer et al. (2021) and Rau et al. (2020):

– The measured well water levels were de-trended using a moving linear regression filter with a 3-day window (Fig. 1d)
and the results are shown in Fig. A2.

– Amplitudes and phases of ten tidal harmonic constituents were jointly estimated using *Harmonic Least Squares* (HALS) (Fig. 1e).

– From HALS, amplitudes and phases of the $M_2$ and $S_2$ tidal components were obtained for the Earth tide strains (Fig. 4a), barometric pressure (Fig. 4b) and hydraulic heads (Fig. 4c,d).

– Complete disentanglement of the groundwater response to Earth and atmospheric tide influences was done for $S_2$ following the method established by Rau et al. (2020) (Fig. 1f).

The resulting amplitude of the hydraulic head (abbreviated as $GW$ for groundwater) $A^{GW}$ can be divided by the Earth tide strain amplitude (abbreviated as $ETP$ for Earth tides) $A^{ETP}$, to obtain the amplitude ratio (Fig. 4e)

$$A_o^{ET} = \frac{A^{GW}}{A^{ETP}}. \tag{15}$$

The phase shift $\Delta\phi_o^{ET}$, can be obtained as the difference between the obtained phase of the hydraulic head measurements $\phi^{GW}$ and the computed Earth tide strain prediction, $\phi^{ETP}$ as

$$\Delta\phi_o^{ET} = \phi^{GW} - \phi^{ETP}. \tag{16}$$

Resulting $A_o^{ET}$ and $\Delta\phi_o^{ET}$ for hydraulic head and areal Earth tide strain for borehole B1 and B2 are presented in Fig. 4e and Table A1.



**Figure 2.** (a) Map of the study site, including surface water features, borehole locations and, location of the transect from A-B, (b) transect showing simplified geology adapted from (Tickell, 2017) and (c) lithological logs from both studied boreholes. $DD$ stands for decimal degrees.

Analogously, the ratio between the resulting amplitude of HALS of the hydraulic head $A^{GW}$ can be divided by the measured (time series) barometric pressure (abbreviated as $ATP$ for atmospheric tides) $A_{ATP}$ to obtain the amplitude ratio

$$A_o^{AT} = \frac{A^{GW}}{A^{ATP}} \rho g. \tag{17}$$



**Figure 3.** Time series of: (a) computed Earth tide strain in nano-strain (nstr), (b) measurements of barometric pressure, and hydraulic head time series measured in boreholes B1 (c) and B2 (d).

The phase shift $\Delta\phi_o^{AT}$, can be obtained as the difference between the obtained phase of the hydraulic head measurements $\phi^{GW}$ and the measured barometric pressure, $\phi^{ATP}$ as

$$\Delta\phi_o^{AT} = \phi^{GW} - \phi^{ATP}. \tag{18}$$

Resulting $A_o^{AT}$ and $\Delta\phi_o^{AT}$ for hydraulic head and areal Earth tide strain for borehole B1 and B2 are presented in Fig. 4f and Table A2.



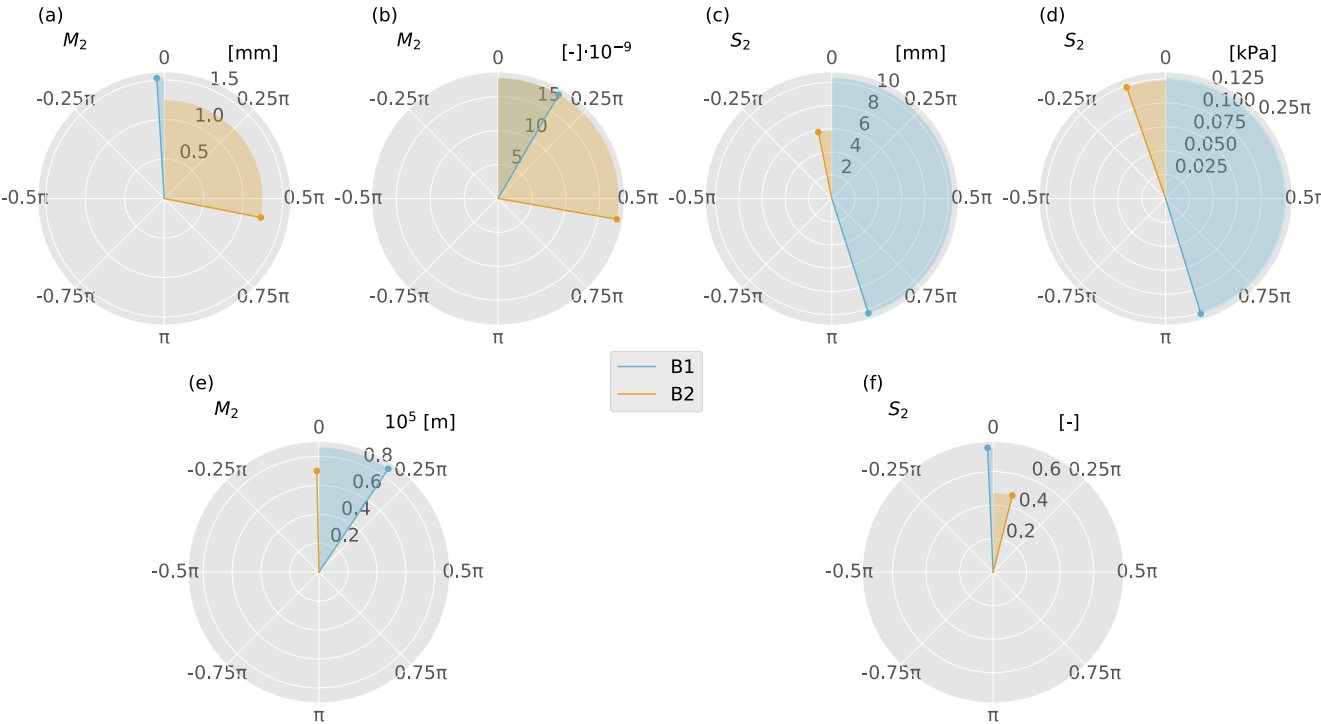

**Figure 4.** Polar plots showing the $M_2$ and $S_2$ harmonics estimated from hydraulic heads in response to Earth and atmospheric tides for boreholes B1 and B2. (a) $M_2$ constituent in measured well water levels. (b) $M_2$ constituent in Earth tide strain data calculated at well locations. (c) $S_2$ constituent in well water levels. (d) $S_2$ constituent in Earth tide strain data calculated at well locations. (e) Amplitude and phase shift of the $M_2$ constituent, equations 15 and 16. (f) Amplitude ratio and phase shift of the $S_2$ constituent, equations 17 and 18 .

### 3.3 Estimation of subsurface properties

To estimate subsurface parameters from the groundwater response to Earth tides, the analytical solution by Wang et al. (2018) was fitted to the $M_2$ harmonic component extracted from field data. This analytical describes the well water level fluctuations, $h_{w,o}^{ET}$, caused by the harmonic compression of the subsurface from Earth tides (abbreviated as $ET$).

The reduction in amplitude of an harmonic signal is described by the ratio between the far field pressure generated by Earth tide strain and the fluid level in the borehole and is known as amplitude ratio(Hsieh et al., 1987)

$$A^{ET} = \frac{h_{w,o}^{ET}}{\epsilon}, \tag{19}$$

where $\epsilon$ is the unit-less areal strain. The time lag between the far field pressure and the fluid level in the borehole is known as phase shift Hsieh et al. (1987)

$$\Delta\phi^{ET} = arg\left(\frac{h_{w,o}^{ET} S_\epsilon}{\epsilon}\right). \tag{20}$$




In theory, the obtained amplitude and phase shift from field measurements (equations 15 and 16) should be the same as those obtained using the analytical solution (Eq. 19 and 20). Since the observed amplitude, $A_o^{ET}$, and phase shift, $\Delta\phi_o^{ET}$, are measurable in the field, they can be used to fit parameters of the analytical solution of Wang et al. (2018) with a non-linear solver to find roots (Fig. 1g). To do so, the following objective function has to be minimised

$$OF^{ET} = \left| \frac{A_o^{ET} - A^{ET}}{A_o^{ET}} \right| + \left| \frac{\Delta\phi_o^{ET} - \Delta\phi^{ET}}{\Delta\phi_o^{ET}} \right|. \tag{21}$$

Since phase shifts can be orders of magnitude grater than amplitude ratio, $OF^{ET}$, is normalised to avoid that one term dominates the solution. Assuming that the borehole construction parameters are known ($H_a$, $H_l$, $r_c$ and $r_w$), three parameters can be estimated, i.e., hydraulic conductivity of the aquifer ($k_a$), vertical hydraulic conductivity of the leaky layer ($k_l$), and specific storage at constant strain ($S_\epsilon$). Once $S_\epsilon$ is obtained, effective porosity can be computed if the material is unconsolidated using (Cheng, 2016; Verruijt, 2013; Wang, 2017)

$$n = \frac{S_\epsilon K_f}{\rho g}, \tag{22}$$

where $K_f$ is the bulk modulus of the fluid ($K_f = 2.2 \cdot 10^9 \ Pa$ for freshwater).

In analogy to Earth tides, the field measurements of barometric pressure and well water levels in the field should match the those obtained by analytical methods. Thus, the obtained amplitude ratio, $A_o^{AT}$, and phase shift, $\Delta\phi_o^{AT}$, in the field (computed later with Eq. 17 and 18, respectively) can be used to estimate subsurface parameters by iterative non-linear numerical methods (Fig. 1h). The function to minimise is

$$OF^{AT} = \left| \frac{A_o^{AT} - A^{AT}}{A_o^{AT}} \right| + \left| \frac{\Delta\phi_o^{AT} - \Delta\phi^{AT}}{\Delta\phi_o^{AT}} \right|. \tag{23}$$

The non-linear search allows for the iterative fitting of four parameters: hydraulic conductivity of the aquifer ($k_a$), vertical hydraulic conductivity of the leaky layer ($k_l$), bulk modulus ($K$), and specific storage at constant strain ($S_\epsilon$). Additionally, specific storage at constant stress ($S_\sigma$) can be estimated using Eq. 11.

Once $S_\epsilon$ is estimated, porosity can be computed with Eq. 22. If values of specific storage, $S$ are known (i.e., from a different characterisation method such as pumping tests), then shear modulus can also be estimated as (Cheng, 2016; Verruijt, 2013; Wang, 2017)

$$G = \frac{3}{4} \frac{(1 - K(S - S_\epsilon/\rho g))}{S - S_\epsilon/\rho g}. \tag{24}$$

By effectively combining hydraulic and poroelastic theory, this approach expands the number of parameters that can be estimated.

By solving Eq. 21, aquifer hydraulic conductivity $k_a$, specific storage at constant strain $S_\epsilon$, and vertical hydraulic conductivity of the aquitard $k_l$ can be estimated. Eq. 23 allows estimation of aquifer hydraulic conductivity $k_a$, specific storage at constant strain $S_\sigma$, vertical hydraulic conductivity of the aquitard $k_l$, and bulk modulus $K$. Once specific storage at constant strain is quantified, porosity $n$, can be estimated with Eq. 22. If the specific storage is known, shear modulus $G$, can be estimated with equation Eq. 24. Equations 21 and 23 can be solved using non-linear iteration (Fig. 1g and 1h).





The non-linear inversion was performed in two steps to help the iterative method converge to a global minimum instead of a local one. Firstly, the solution space of the objective function was divided into intervals within feasible ranges of subsurface properties, creating a feasible objective space, thus bounding the initial conditions for the least-squares algorithm. Secondly, 1,000 randomly generated values following a log-normal distribution were fed as initial conditions to the least-squares algorithm, the array of parameters that converges to the best fit among them, was considered to be the global minimum of the non-linear search (Aster et al., 2018).

## 3.4 Hydraulic and geomechanical properties

Values from Earth tide analysis and atmospheric tide analysis are presented in Tables 2 and 3. Further, the estimated aquifer hydraulic conductivity, specific storage at constant strain and aquitard vertical hydraulic conductivity for boreholes B1 and B2 are shown Fig. 5.

**Table 2.** Estimated subsurface parameters from Earth tide analysis.

| | Non-linear search results | | | |
|---|---|---|---|---|
| Borehole | $k_a$ $[ms^{-1}]$ | $S_\epsilon$ $[m^{-1}]$ | $k_l$ $[ms^{-1}]$ | n [-] |
| B1 | $1.1 \cdot 10^{-5}$ | $1.8 \cdot 10^{-6}$ | $5.4 \cdot 10^{-8}$ | 0.37 |
| B2 | $1.0 \cdot 10^{-4}$ | $3.8 \cdot 10^{-7}$ | $1.1 \cdot 10^{-8}$ | 0.08 |

**Table 3.** Estimated subsurface parameters from atmospheric tide analysis.

| | Non-linear search results | | | | | |
|---|---|---|---|---|---|---|
| Borehole | $k_a$ $[ms^{-1}]$ | $S_\epsilon$ $[m^{-1}]$ | $k_l$ $[ms^{-1}]$ | K $[GPa]$ | $S_\sigma$ $[m^{-1}]$ | n [-] |
| B1 | $1.6 \cdot 10^{-5}$ | $1.8 \cdot 10^{-6}$ | $8.0 \cdot 10^{-10}$ | 0.3 | $3.5 \cdot 10^{-5}$ | 0.4 |
| B2 | $1.0 \cdot 10^{-4}$ | $5.0 \cdot 10^{-7}$ | $6.0 \cdot 10^{-8}$ | 10.0 | $1.5 \cdot 10^{-6}$ | 0.11 |

The basic assumption of undrained conditions applies to the analytical solutions by Wang et al. (2018) and this study (Eq. 8). To assess whether this condition is fulfilled, both Earth and atmospheric tide analyses, were assessed separately:

1. For Earth tide analysis, Bastias et al. (2022) numerically computed the level of drainage over depth for different arrays of subsurface properties. Despite the estimated $k_l$ being outside the range presented by Bastias et al. (2022), it can be extrapolated. At borehole B1, the aquifer is within undrained conditions. At borehole B2, it is within the transition zone between drained and undrained.





2. For atmospheric tide analysis, Wang (2017) defined the depth of undrained conditions as

$$\delta = \sqrt{\frac{2c}{\omega}}, \tag{25}$$

where $c$ is the consolidation coefficient. For boreholes B1 and B2, undrained conditions are found at depths higher than $2.3$ and $40.6\ m$, respectively, under atmospheric tide loading.

Consequently, for the estimated parameters in this study, B2 borders drained conditions, and the generated confined pore pressure by tidal forcing is being diminished. This may influence the estimated properties.

The aquifer hydraulic conductivity estimated with PSC complies with previous values of poorly consolidated sands and gravel aquifers in the literature ($5 \cdot 10^{-6} \le k_a \le 10^{-3}\ ms^{-1}$) (Freeze and Cherry, 1979; Tickell, 2017) (Fig. 5a, Tables 2 and 3). Note that the estimated value of $k_a$ is lower compared to the pumping tests in the study site (Table B1). Bastias et al. (2022) studied the area of influence of PSC and concluded that PSC is a small-scale characterisation technique where parameters are estimated in the vicinity of the well screen. This might explain the difference between values presented in this study and the ones obtained with pumping tests (Appendix B), as estimates parameters with small-scale methods will tend to give much lower values than obtained from a full-well or packer pumping test, because small-scale analyses may miss the most permeable intervals that make the greatest contribution to the transmissivity (Maliva, 2016). This idea is supported by previous studies that reported several orders of magnitude differences between traditional hydraulic characterisation methods and PSC (Allègre et al., 2016; Zhang et al., 2019; Valois et al., 2022; Qi et al., 2023). The difference was attributed to issues such as the borehole skin effect (Zhang et al., 2019; Valois et al., 2022) and differing model assumptions (Qi et al., 2023). Furthermore, Zhang et al. (2021) showed good agreement between hydraulic parameters of a consolidated subsurface system derived using PSC and laboratory measurements. This supports our observation that PSC results are representative of a smaller sample volume close to the well screen. However, determining the extent of the area around the well screen affected by flow from tidal forces is outside the scope of this work and requires further investigation. Additionally, reconciling the properties derived from both active and passive approaches will require more research.

The estimated values of specific storage at constant strain for B1 are within the previously reported values in the literature for sand aquifers, $1.13 \cdot 10^{-6} \le S_\epsilon \le 2.27 \cdot 10^{-6}\ m^{-1}$ (Freeze and Cherry, 1979) (Fig. 5b). Porosity, computed with Eq. 22, is also within the reported range, $0.25 \le n \le 0.5$ (Freeze and Cherry, 1979). Conversely, borehole B2 shows values of specific storage at constant strain and porosity are below the expected range Tables 2 and 3. There are several potential causes for this, such as the presence of flow paths that create undrained conditions, leading to a reduction in the generated confined pore pressure and exposing the limitations of passive methods for this borehole. Furthermore, the degree of aquifer consolidation is limited, and the length of the well screen is not representative of the full depth of the aquifer. These factors were not explored in this study and should be the focus of future numerical investigations to better understand their effects on the results.

The estimated aquifer bulk modulus values (Table 3) were consistent with literature values for sands and gravels, typically between $5 \cdot 10^{-2}\ GPa$ and $3 \cdot 10^1\ GPa$ (Das and Das, 2008; Look, 2007). If it is assumed that the average variability of the hydraulic properties in the aquifer is low (Fig. B1b), the values presented in Table B1 (wells W7, W8, W9 and W10) can be used to estimate shear modulus using Eq. 24. Estimated shear modulus values were $0.7$ and $0.03\ GPa$ for B1 and B2





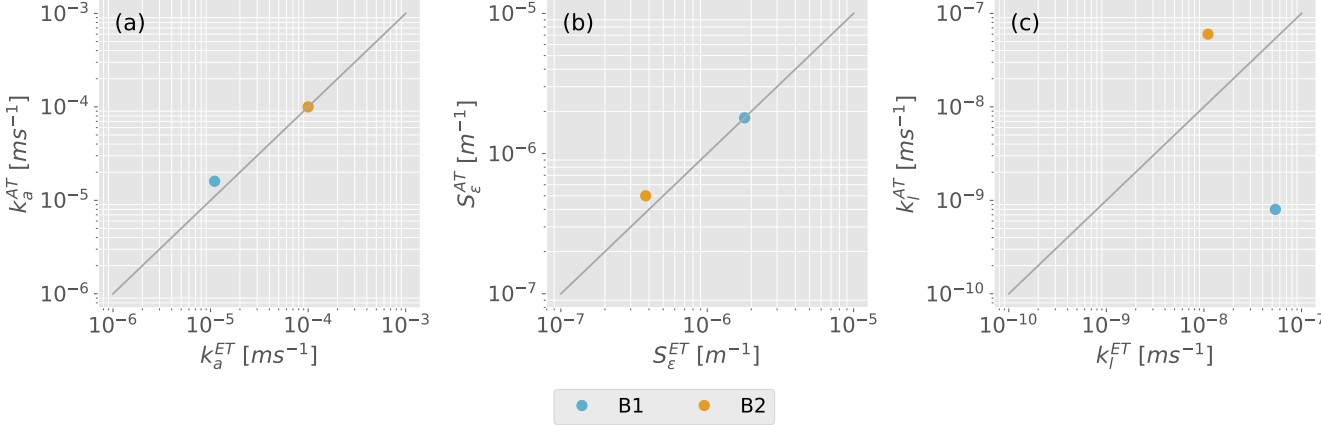

**Figure 5.** Comparison of the subsurface parameters estimated independently using the well water level response to Earth tides and atmospheric tides: (a) hydraulic conductivity of the aquifer, (b) specific storage at constant strain, (c) vertical hydraulic conductivity of the aquitard.

respectively. We note that these values are consistent with expectations reported in the literature for similar lithological settings,
e.g., typically between $8 \cdot 10^{-3} \ GPa$ and $9 \cdot 10^3 \ GPa$ (Das and Das, 2008; Look, 2007).

Compared to the previous analytical solution presented by Rojstaczer and Riley (1990), which describes flow to wells due to barometric loading, the derived analytical solution in this study simplifies the pore pressure wave generated in the vadose zone by assuming that only small vertical flow occurs in the confined layer. Moreover, the solution of Rojstaczer and Riley (1990) requires knowledge of vadose zone properties that are difficult to determine. Furthermore, the continuity equation is solved in
terms of the mean stress equation, allowing for the estimation of mechanical parameters such as bulk modulus and specific storage at constant stress. As shown in our work, this extends the current range of parameters that can be estimated passively (McMillan et al., 2019).

While we present a new analytical solution, we are unable to compare or validate geomechanical results due to a lack of independent measurements. Additionally, the literature comparing subsurface properties using PSC from different methods is
sparse and contains somewhat conflicting conclusions (Allègre et al., 2016; Zhang et al., 2019; Valois et al., 2022; Qi et al., 2023; Zhang et al., 2021). This is likely due to the fact that subsurface investigations often focus on determining hydraulic properties such as hydraulic conductivity and specific storage, which are critical for understanding subsurface fluid flow. Obtaining geomechanical information such as bulk modulus, shear modulus, and stress state can be challenging and may require additional investigation techniques. However, Rau et al. (2022) noted that in-situ stress conditions, stress anisotropy
and scale differences complicate comparisons with laboratory methods. We believe that systematic investigations in different archetypes of formations, including the use of borehole geophysical investigation techniques and careful laboratory testing of material samples, could help to clarify scale and heterogeneity influences, reconcile the different theories, and provide further confidence in values derived from PSC.





## 4    Conclusions

We have introduced a novel analytical solution based on the mean stress flow equation for modelling flow to wells induced by atmospheric loading. We integrate this mean stress solution into a comprehensive workflow for estimating subsurface hydraulic and geomechanical properties using the groundwater response to Earth and atmospheric tides, applied this to a standard groundwater monitoring data set from the Northern Territory (Australia) and discussed the results with hydraulic properties from pumping tests and geomechanical literature values for similar lithological settings. Our new solution allows

estimation of hydraulic conductivity of the aquifer, vertical hydraulic conductivity of the aquitard, porosity, specific storage at constant strain, specific storage at constant stress and bulk modulus. The advantages are estimation of additional subsurface properties without the need for knowledge of vadose zone properties.

We compared the hydraulic properties estimated independently using the groundwater response to Earth tides and atmospheric pressure. The estimated values of aquifer hydraulic conductivity with Earth tidal analysis were $1.1 \cdot 10^{-5}\ ms^{-1}$ and

$1.1 \cdot 10^{-4}\ ms^{-1}$ for borehole B1 and B2, respectively. Meanwhile, with the mean stress solution, the estimated values of aquifer hydraulic conductivity were $1.6 \cdot 10^{-5}\ ms^{-1}$ and $1.0 \cdot 10^{-4}\ ms^{-1}$ for borehole B1 and B2, respectively. These estimated values were lower than those estimated using pumping tests for the region between Mary River National Park and Kakadu National Park (ranging from $6 \cdot 10^{-4}$ to $8 \cdot 10^{-5}\ ms^{-1}$). This difference is consistent with the literature and supports the idea that PSC is a small-scale characterisation method.

The estimated specific storage at constant strain for borehole B2 was $3.8 \cdot 10^{-7}$ and $5.0 \cdot 10^{-7}\ m^{-1}$ with Earth tidal analysis and the mean stress equation, respectively. This indicates that the response near borehole B2 is drained since the estimated values are lower than the reported bounds in the literature. Consequently, the drained conditions reduce the confined pore pressure generated by tides. The estimated values of aquitard vertical hydraulic conductivity differed from the pumping tests by orders of magnitude but suggest that the aquifer in both boreholes is semi-confined with small leakage.

The bulk and shear moduli aligned with literature values for the formation type, confirming that PSC has the potential to enhance field investigations. However, for PSC to be applied successfully, it is necessary for the basic physical assumptions underlying the analytical solutions to be valid. This can be challenging to determine in situations such as confined and undrained hydraulic conditions or an unconsolidated system where the Biot coefficient is unknown. As a result, PSC can only be applied in hydrogeological settings that adhere to the theoretical framework.

Compared to established methods like hydraulic testing, using PSC requires a better understanding of hydraulic and hydro-geomechanical theory as well as signal processing. However, PSC is less costly and effort-intensive because it only requires monitoring datasets that typically meet standard practice criteria. The literature reflects confusion about the suitability of theory and a lack of geomechanical testing alongside hydraulic testing, making it challenging to validate poroelastic properties. Systematic investigations involving a range of archetypal formations with a combination of hydraulic, geophysical, geotechnical

field and laboratory tests are needed to validate PSC. This would help compare properties from rigid and elastic formations, reconcile theories, and support groundwater and geotechnical investigations.



*Code availability.* Python based code which streamlines the data analysis and subsurface properties estimation are provided after submission.

*Data availability.* The datasets are provided to editors and reviewers during the review process and to the public should this work be accepted.

*Code and data availability.* The code is provided to editors and reviewers during the review process and to the public should this work be
accepted.

**Appendix A**

The hydraulic head measurements in borehole B1 and B2 are shown in Fig. A1. Outliers were detected and eliminated with the procedure described in Section 3.2.

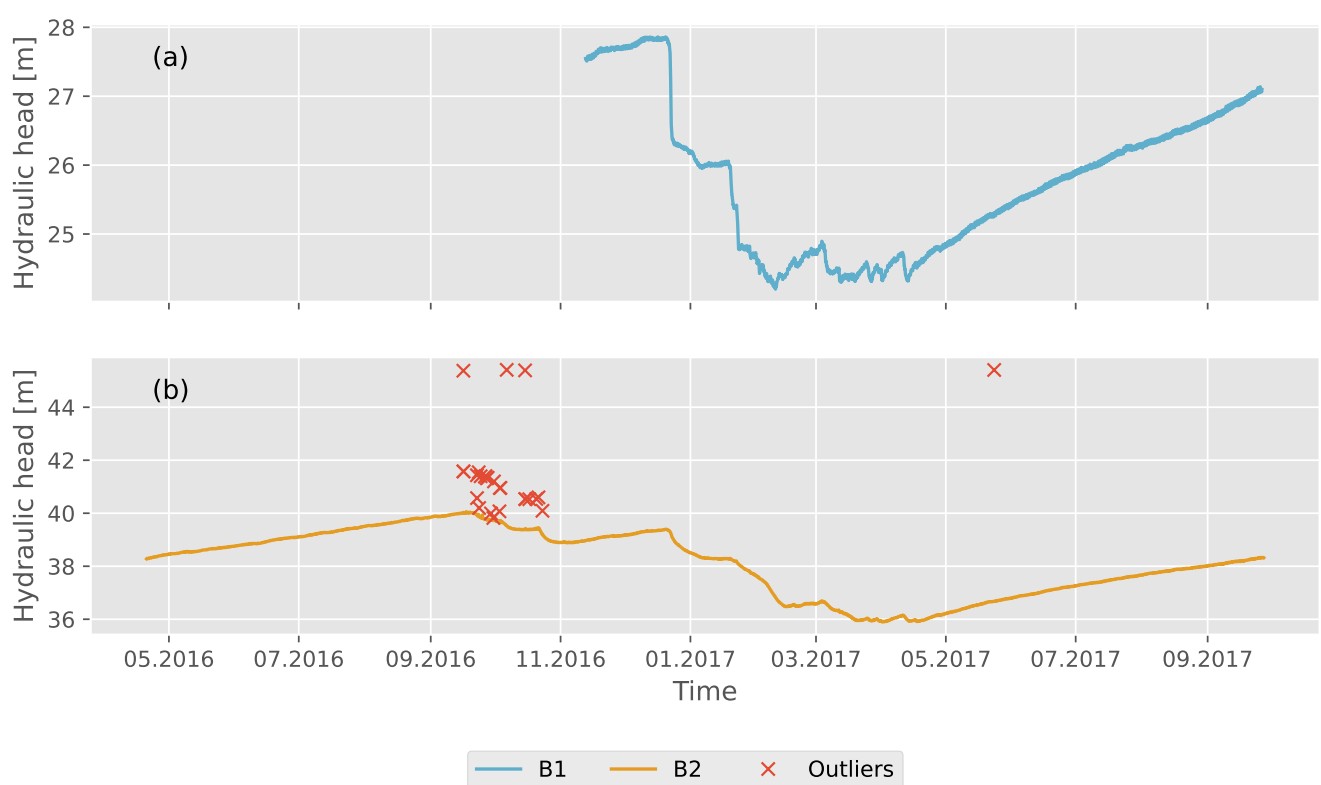

**Figure A1.** Hydraulic head time series and outliers measured in boreholes (a) B1 and (b) B2.



Computed areal Earth tide strain, measured barometric pressure and hydraulic head of borehole B1 and B2 were de-trended
using a moving meadian filter with 3-day window, Sec. 3.2 and Fig. A2.



**Figure A2.** The corresponding de-trended time series showing only components with frequencies up to 3 cpd; (a) computed Earth strain, (b) measured atmospheric pressure, hydraulic head (c) B1 and (d) B2.

Harmonic constituents were obtained applying harmonic least squares (HALS), results of amplitude and phase shift to the $M_2$ signal are shown in Table A1. Analogously, the amplitude and phase shift to the $S_2$ signal are shown in Table A2.

**Table A1.** Amplitude ratio and phase shift obtained with HALS for the $M_2$ constituent.

| Borehole | $A^{ETP}$ [-] | $\Delta\phi^{ETP}$ [°] | $A^{GW}$ [m] | $\Delta\phi^{GW}$ [°] | $A_o^{ET}$ [m] | $\Delta\phi_o^{ET}$ [°] |
|---|---|---|---|---|---|---|
| B1 | $26.57 \cdot 10^{-9}$ | 0.52 | 0.0015 | $-0.061$ | 57386.49 | 0.59 |
| B2 | $26.65 \cdot 10^{-9}$ | 1.74 | 0.0012 | 1.76 | 46710.38 | $-0.02$ |





**Table A2.** Amplitude ratio and phase shift obtained with HALS for the $S_2$ constituent.

| Borehole | $A^{ATP}$ [$kPa$] | $\Delta\phi^{ATP}$ [°] | $A^{GW}$ [$m$] | $\Delta\phi^{GW}$ [°] | $A_o^{AT}$ [-] | $\Delta\phi_o^{AT}$ [°] |
|---|---|---|---|---|---|---|
| B1 | 0.12 | 2.84 | $1.0 \cdot 10^{-2}$ | 2.83 | 0.82 | $-0.71$ |
| B2 | 0.12 | $-0.33$ | $0.58 \cdot 10^{-3}$ | $-0.20$ | 0.47 | 7.83 |



**Appendix B**

Time–drawdown data from five two-well pumping tests in the Mary–Wildman rivers area were reinterpreted using appropriate
drawdown solutions using a two-step process (see Table B1 and W1 to W10 in Figure B1a) (Turnadge et al., 2018). The
time–drawdown data, were used to identify appropriate pumping test analysis solutions. These included: the solutions of
Barker (1988) for fractured rock flow under confined conditions; Hantush (1960) for leaky conditions; and Neuman (1974) for
unconfined conditions.

**Figure B1.** Map of the study site. (a) shows PSC boreholes, barometric sensor and location of the wells were pumping tests were performed
(wells W1 to W10). (b) shows the surface geology of the Mesozoic/Cenozoic strata (modified from NT Geological Survey digital data,
(Tickell, 2017))



**Table B1.** Details of five historical two-well pumping tests undertaken in the Mary–Wildman rivers area, including aquifer types interpreted from time–drawdown responses and forward solutions used to estimate aquifer hydraulic properties via model inversion. Hydraulic property values are displayed with root mean square error of optimised least squares solution. For each estimated parameter, the optimal value derived from least squares estimation is provided, as well as the approximate 95 % confidence interval.

| Production well ID | | Observation well ID | Attributed aquifer | Confinement type | Inversion solution | RMSE [m] | $k_a \cdot 10^{-4}$ [$m\,s^{-1}$] | $S_s \cdot 10^{-5}$ [$m^{-1}$] | $S_y \cdot 10^{-3}$ [-] |
|---|---|---|---|---|---|---|---|---|---|
| W1 RN023158 | - | W2 RN023230 | Cretaceous sand | Unconfined | Neuman (1974) | $1 \cdot 10^{-2}$ | $4.1 \pm 0.3$ | N/A | $1.0 \pm 0.4$ |
| W3 RN024668 | - | W4 RN024596 | Wildman Siltstone | Confined | Barker (1988) | $9 \cdot 10^{-4}$ | $4.6 \pm 0.9$ | $8.0 \pm 0.6$ | N/A |
| W5 RN024669 | - | W6 RN024228 | Wildman Siltstone | Confined | Barker (1988) | $2 \cdot 10^{-3}$ | $0.8 \pm 0.2$ | $1.0 \pm 0.8$ | N/A |
| W7 RN024763 | - | W8 RN024764 | Cretaceous sand | Leaky | Hantush (1960) | $1 \cdot 10^{-4}$ | $6.3 \pm 0.3$ | $5.0 \pm 0.5$ | N/A |
| W9 RN039769 | - | W10 RN039768 | Cretaceous sand | Leaky | Hantush (1960) | $1 \cdot 10^{-3}$ | $4.1 \pm 0.4$ | $7.0 \pm 0.8$ | N/A |



*Author contributions.* JMBE analysed the datasets, created the figures and wrote the first manuscript draft. CT and RC provided the datasets
and reviewed the work. CT analysed the pumping tests. PB reviewed the manuscript and provided conceptual suggestions. GCR obtained the
funding, supervised JMBE and actively contributed to all aspects of this work.

*Competing interests.* The authors declare that they have no competing interests.

*Acknowledgements.* This project has received funding from the German Research Council (DFG) grant agreement number 424795466. The
authors would like to thank the following individuals who conducted field work to collect datasets used here: Steven Tickell, Ursula Zaar,
Gary Willis, Steve Dwyer, Emma Jackson, Ian Bate, Stanley Smith, Alec Deslandes, Karen Barry, Simone Gelsinari, and Julia Knollmann.



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
