# Peer review of "Technical note: Novel analytical solution for groundwater response to atmospheric tides"

_EGUsphere, 2023_

## Referee Comment (RC1)

**Review of:** Technical note: Novel analytical solution for groundwater response to atmospheric tides.

The authors present a new analytical solution for modelling flow between a subsurface-well system caused by harmonic atmospheric loading. They integrate this into a comprehensive workflow that also estimates subsurface properties using a well-established Earth tide method. The method is applied to two wells in Australia to estimate hydraulics parameters and compare with the results calculated in the literature.

**General comments:**
1. Uncertainties of the calculated hydraulic parameters are missing.
   (a) HALS in Figure 4 showed the best value, is there any consideration of uncertainty or range of error?
   (b) Table 2, Table 3, Table A1 and Table A2 do not take into account the uncertainty and error range values, these values should be presented as a range value.
   (c) Figure5 calculation results do not reflect the error range.
2. Oceanic tides could have a large influence on the results. Amplitudes (and phases) can be computed using SPOTL to check whether they are negligible or not.
3. In section 3.2, the ET parameters are calculated using M2 and S2. S2 waves are affected not only by earth tide but also by barometric pressure. Is the effect of barometric pressure on the Earth tide response considered when using S2 waves to calculate Earth tide, and is its effect on the tidal response removed?
4. There is Biot coefficient $\alpha$ in equation 1, but the solution equation 8 does not contain the Biot coefficient, so is the biotech coefficient eliminated in the calculation or is it assumed to be 1 to represent the consolidation system? Is mean stress solution the method to find the parameters of the consolidation system?

5. In some articles(Acworth *et al.*, 2016; McMillan *et al.*, 2021), porosity can be calculated by this equation $n = \frac{S_s K_f}{\rho g} BE$, compared to equation 33 $n = \frac{S_\epsilon K_f}{\rho g}$,of the article, is it possible to get the relationship between specific storage at constant strain and specific storage is $S_\epsilon = S_s * BE$ ? then Equation 24 and the calculated G can be further improved.

**Minnor comments:**
1. The statement in L28-31 is not accurate. The tidal component of O1 is also very large, S1 is not a major component of Earth tide. It's a little weird to mention S1 here.
2. L37-39: Negative skin effects could also be associated to positive phase lag (Valois et al, 2022).

   Valois, R., Rau, G. C., Vouillamoz, J. M., & Derode, B. (2022). Estimating hydraulic properties of the shallow subsurface using the groundwater response to Earth and atmospheric tides: a comparison with pumping tests. Water Resources Research, 58(5), e2021WR031666.

3. What is the t in equation 1?
4. L86 should be changed to "r is the radius".
5. L176-179: What is the sampling frequency of water level and barometric pressure sensor

records, and what is the time interval used for air barometric, water level, and strain data?

6.  The title of Figure 2(b) should emphasize that it is only the geological unit between the AB, as well B1 is not represented in this figure.

7.  What does $\rho$ and $g$ stand for in equation 17 and 22, and what are the values used for each?

8.  The method description in Section3.2 is not very clear, it is not very clear what wave components are used in which method, it should be explained. And what is the study time period used for AT and ET respectively should be explained.

9.  What is the black line in Figure 5? It should be reflected in the title of the figure.

10. L296-297 need to add references.

11. It needs to be clarified whether the values calculated in L334-336 are the best fitting parameters.

12. The conclusion should include the proposal of future research.

---

## Author Comment (AC1)

**Response to Reviewer Comments 1**

We thank the reviewer for the comments. Note that we use the abbreviation **RC** for instance refer to reviewer comments 1, **EC** for editor and **AR** for our authors' response in the following. To highlight the nature of our reply, we use two colours; with green for agreement and yellow for misunderstanding. We explain all our reasoning and changes made to the manuscript. Removed text is shown in red, e.g., this text has been removed. New text is shown in blue, e.g., this text has been added.

**RC:** The authors present a new analytical solution for modelling flow between a subsurface-well system caused by harmonic atmospheric loading. They integrate this into a comprehensive workflow that also estimates subsurface properties using a well-established Earth tide method. The method is applied to two wells in Australia to estimate hydraulics parameters and compare with the results calculated in the literature.

**AR:** We appreciate the time and effort in evaluating the manuscript and providing such positive feedback to improve our work. Please find a detailed response to every comment below.

RC: Uncertainties of the calculated hydraulic parameters are missing.

- 1. HALS in Figure 4 showed the best value, is there any consideration of uncertainty or range of error?
- 2. Table 2, Table 3, Table A1 and Table A2 do not take into account the uncertainty and error range values, these values should be presented as a range value.
- 3. Figure5 calculation results do not reflect the error range.

**AR:** We partially agree: Uncertainty analysis is crucial for assessing the resilience of the solution obtained through inverse analytical methods. Nevertheless, it should be noted that this process relies on a series of steps, each involving non-linear relationships, which significantly complicates the propagation of uncertainty. Given that the primary objective of this current study is the development and demonstration of a novel analytical solution (technical note), the estimation of uncertainties will be deferred to more comprehensive applications in future investigations.

**RC:** Oceanic tides could have a large influence on the results. Amplitudes (and phases) can be computed using SPOTL to check whether they are negligible or not.

**AR:** We agree: We computed the Ocean tide loading by OLMPP by H G Scherneck, Onsala Space Observatory, 2023, at the study site (Lat: -12.607700 DD and Long: 131.829500 DD). The following table summarises the amplitudes of the computed displacements for both, Ocean Tides and Earth tides. The ratio of magnitude between the harmonic constitutive computed as (Earth tide)c/(Ocean tide)c, where c represents a given frequency, are 21.662 for the  $M_2$  and 37.770 for the  $S_2$ , which shows that the impact of Ocean Tides is insignificant.

|                | 1           | 42          | 32          |             |  |  |
|----------------|-------------|-------------|-------------|-------------|--|--|
| Amplitudes [m] | Earth tides | Ocean tides | Earth tides | Ocean tides |  |  |
| Radial         | 0.13678     | 0.00575     | 0.06556     | 0.00144     |  |  |
| Tangent EW     | 0.00687     | 0.00192     | 0.00330     | 0.00063     |  |  |
| Tangent NS     | 0.00687     | 0.00184     | 0.00330     | 0.00076     |  |  |
| Magnitude      | 0.13712     | 0.00633     | 0.06572     | 0.00174     |  |  |
| Ratio          | 21.         | .662        | 37.770      |             |  |  |

Table 1: Computed displacement generated by Ocean and Earth tidal loading.

Suggested revision at line 185: Earth tide strains were calculated using *PyGTide* (Rau, 2018) which is based on the widely used ETERNA PREDICT software Wenzel (1996) (Fig. 1c). To ensure that ocean tides have a negligible influence on the strain, the ocean tidal effect was computed using the Ocean Loading Model for Permanent and Persistent (OLMPP) method developed by H. G. Scherneck at the Onsala Space Observatory in 2023, at the study site located at latitude -12.607700 DD and longitude 131.829500 DD. For the  $M_2$  Earth tides, the amplitude is 21 times higher than that of the ocean loading, while for the  $S_2$  Earth tides, the amplitude is 37 times higher.

**RC:** In section 3.2, the ET parameters are calculated using M2 and S2. S2 waves are affected not only by earth tide but also by barometric pressure. Is the effect of barometric pressure on the Earth tide response considered when using S2 waves to calculate Earth tide, and is its effect on the tidal response removed?

**AR:** We agree: We separated the effects of Earth tides and atmospheric tides for the  $S_2$  tidal component. Although this signal disentanglement was described in the methodology, it seems that our explanation was not clear enough.

Suggested revision at line 195: Complete disentanglement of the groundwater response to Earth and atmospheric tide influences was done for *S*2 following the method established by Rau et al. (2020) (Fig. 1f).

Given that the  $S_2$  tidal constituent comprises both Earth tidal and atmospheric tide influences, it is necessary to perform a separation of these effects. We quantitatively disentangled the groundwater response to the influences of both Earth and atmospheric tides for the  $S_2$  component in the frequency domain. The approach outlined by Rau et al. (2020b) (Fig. 1f)effectively separates the  $S_2$  harmonic embedded in groundwater into two harmonics that can each be attributed either to the associated Earth or atmospheric tide.

**RC:** There is Biot coefficient in equation 1, but the solution equation 8 does not contain the Biot coefficient, so is the biotech coefficient eliminated in the calculation or is it assumed to be 1 to represent the consolidation system? Is mean stress solution the method to find the parameters of the consolidation system?

**AR: We agree** The Biot coefficient is assumed to be one since the subsurface system is unconsolidated (i.e. sands, gravels and clays). If the system was consolidated (i.e. different types of rocks), then the Biot Coefficient would typically vary between  $n < \alpha < 1$ , where *n* is porosity. Then the confined pore pressure generated by atmospheric tides is given by,

$$p_f = R\left(-\frac{\alpha}{K}\sigma + \xi\right),\tag{1}$$

where  $p_f$  is the fluid (i.e., water in this study) pore pressure;  $\sigma$  is the mean stress; *K* is the drained bulk modulus of the solid material;  $\xi$  is the change in fluid content, can be used to quantify changes in pore pressure resulting from hydraulic gradients (Cheng, 2016; Verruijt, 2013; Wang, 2017). The sign of this parameter indicates the direction of fluid flow, whether it is leaving or entering a specific porous medium. In our analysis, we consider undrained conditions ( $\xi = 0$ ) and solely unconsolidated systems ( $\alpha = 1$ ). In future investigations, consolidated systems where  $\alpha \neq 1$  could be incorporated. However, in such cases, fitting the analytical solution during the inversion process would necessitate the inclusion of a fifth parameter, introducing greater non-linearity to the equation and rendering the fitting task even more challenging.

Suggested revision at line 114: Assuming undrained conditions ( $\xi = 0$ ) and an unconsolidated system ( $\alpha = 1$ ), we solved Eq. 1 for steady-state conditions to obtain the periodic water level in an open borehole  $h_w^{AT} = h_{w,o}^{AT} e^{i\omega t}$  caused by atmospheric loading [...]

**RC:** In some articles(Acworth et al., 2016; McMillan et al., 2021), porosity can be calculated by this equation  $n = \frac{S_s K_f}{\rho_g} BE$ , compared to equation  $33 n = \frac{S_{\varepsilon} K_f}{\rho_g} BE$ , of the article, is it possible to get the relationship between specific storage at constant strain and specific storage is  $S_{\varepsilon} = S_s BE$ ? then Equation 24 and the calculated *G* can be further improved.

**AR:** We agree, but point out that this only works for unconsolidated systems (i.e., the compressibility of the grains is much lower than that of the bulk). In that case:

$$\gamma = \frac{1}{K_{\nu}S} \tag{2}$$

and

$$\gamma = \frac{1}{K_v^{(u)} S_\varepsilon},\tag{3}$$

where  $\gamma$  is the tidal efficiency,  $K_{\nu}$  and  $K_{\nu}^{(u)}$  are often called the drained and undrained vertical incompressibility, respectively. Combining Eq. 1 and Eq. 2

$$\frac{S_{\varepsilon}}{S} = \frac{K_{\nu}}{K_{\nu}^{(u)}} \tag{4}$$

and, by definition, Barometric Efficiency (BE) is

$$BE = \frac{K_v}{K_v^{(u)}},\tag{5}$$

hence,

$$S_{\varepsilon} = S \cdot BE. \tag{6}$$

Suggested revision at line 243: Once  $S_{\varepsilon}$  is estimated, porosity can be computed with Eq. 22. If values of specific storage, *S* are known (i.e., from a different characterisation method such as pumping tests), then shear modulus can also be estimated as (Cheng, 2016; Cheng, 2013; Cheng, 2017)

By effectively combining hydraulic and poroelastic theory, this approach expands the number of parameters that can be estimated.

Specific storage, *S*, can be obtianed if the Barometric Efficiency ( $A_o^{AT}$  in this study) is known (Cheng, 2016; Verruijt, 2013; Wang, 2017),

$$S = \frac{S_{\mathcal{E}}}{A_o^{AT}},\tag{7}$$

then shear modulus can also be estimated as (Cheng, 2016; Verruijt, 2013; Wang, 2017)

$$G = \frac{3}{4} \frac{(1 - K(S - S_{\varepsilon}/\rho g))}{S - S_{\varepsilon}/\rho g}.$$
(8)

Suggested revision at line 253: [...] Once specific storage at constant strain is quantified, porosity n, can be estimated with Eq. 22. The specific storage S, can be obtained with Eq. 25 and shear modulus G, can be estimated with equation Eq. 25. [...]

Suggested revision in Table 3:

|          | Tion mour source robuits |                            |                      |         |                      |       |                     |         |
|----------|--------------------------|----------------------------|----------------------|---------|----------------------|-------|---------------------|---------|
| Borehole | $k_a \ [ms^{-1}]$        | $S_{\varepsilon} [m^{-1}]$ | $k_l \ [ms^{-1}]$    | K [GPa] | $S_{\sigma}[m^{-1}]$ | n [-] | $S[m^{-1}]$         | G [GPa] |
| B1       | $1.6 \cdot 10^{-5}$      | $1.8 \cdot 10^{-6}$        | $8.0 \cdot 10^{-10}$ | 0.3     | $3.5 \cdot 10^{-5}$  | 0.4   | $2.2 \cdot 10^{-6}$ | 18.5    |
| B2       | $1.0\cdot10^{-4}$        | $5.0\cdot10^{-7}$          | $6.0\cdot 10^{-8}$   | 8.0     | $1.5\cdot10^{-6}$    | 0.11  | $1.6 \cdot 10^{-6}$ | 0.63    |

Non-linear search results

Suggested revision at line 301: If it is assumed that the average variability of the hydraulic properties in the aquifer is low B1, the values presented in Table B1 (wells W7, W8, W9 and W10) can be used to estimate shear modulus using Eq. 8. Estimated shear modulus values were 0.7 and 0.03 *GPa* for B1 and B2 respectively. The estimated values of specific storage were consistent with the pumping tests performed at the study site (assuming low spacial variability of the hydro geomechanical properties, Fig. B1). Once specific storage is estimated, shear modulus can be estimated with Eq. 8. We note that these values are consistent with expectations reported in the literature for similar lithological settings, e.g., typically between  $8 \cdot 10^{-3}$  *GPa* and  $9 \cdot 10^3$  *GPa* (Das and Das, 2008; Look, 2007).

**RC:** The statement in L28-31 is not accurate. The tidal component of O1 is also very large, S1 is not a major component of Earth tide. It's a little weird to mention S1 here.

**AR:** We agree: This is a mistake, we changed the tidal component S1 for O1.

Suggested revision at line 29: Dominant frequencies present in groundwater pressure measurements are the S1 O1 (1.0 cpd), M2 (1.93 cpd) and the S2 (2.00 cpd).

**RC:** L37-39: Negative skin effects could also be associated to positive phase lag (Valois et al, 2022).

**AR:** We agree and will revise accordingly.

Suggested revision at line 37: Positive phase shifts (i.e., when well water levels respond before subsurface water pressures to Earth tide-induced strain variations) have been linked to vertical connectivity with adjoining hydrostratigraphic units (Roeloffs et al., 1989) and negative skin effects of the observation well (Valois et al., 2022).

**RC:** What is the t in equation 1?

AR: We agree and corrected our text accordingly

Suggested revision at line 86: Here, t represents time;  $\sigma$  is the mean stress; [...]

**RC:** L86 should be changed to "r is the radius"

**AR:** We agree and corrected our text accordingly

Suggested revision at line 86: [...] r is the radius [...]

**RC:** L176-179: What is the sampling frequency of water level and barometric pressure sensor records, and what is the time interval used for air barometric, water level, and strain data?

**AR:** We agree and added this information to our text accordingly

Suggested revision at line 176: Well water levels were monitored hourly between June 2016 and September 2019 in each borehole using InSitu Level TROLL 400 data loggers (InSitu Inc., USA), the sensor's sampling frequency was set to one sample per hour. The measured pressure heads were converted to hydraulic head values by referencing the dips of depth to water level manually to the surveyed top of casing elevations. Concurrently, barometric pressure was recorded from September 2016 to October 2017 using an InSitu BaroTROLL 500 data logger (InSitu Inc., USA), the sensor's sampling frequency was set to one sample per hour.

Suggested revision at line 184: Earth tide strains were calculated using *PyGTide* (Rau, 2018) which is based on the widely used ETERNA PREDICT software Wenzel (1996) (Fig. 1c). The time series were generated with a frequency of one sample per hour.

**RC:** The title of Figure 2(b) should emphasize that it is only the geological unit between the AB, as well B1 is not represented in this figure

**AR:** We agree and we corrected the caption of the Figure 2(b). Also, to avoid confusion, we removed B2 from the figure.

Suggested revision at caption of Figure 2(b):

(b) transect showing simplified geology adapted from (Tickell, 2017) (b) A-B transect displaying a simplified geology adapted from Tickell (2017).

**RC:** What does  $\rho$  and g stand for in equation 17 and 22, and what are the values used for each?

**AR:** We agree  $\rho$  and g where previously defined in line 95. We agree that is necessary to include the assumed values for these constants

Suggested revision at line 95: where g is gravitational acceleration (9.81  $ms^{-2}$ ) and  $\rho$  the fluid density (1000  $kgm^{-3}$ ).

**RC:** The method description in Section 3.2 is not very clear, it is not very clear what wave components are used in which method, it should be explained. And what is the study time period used for AT and ET respectively should be explained.

**AR:** We agree and revised as follows:**

Suggested revision at line 186: Harmonic tidal components of the ten dominant target frequencies between 0.33 and 2.2 cycles per day (cpd) Harmonic tidal components of the dominant tidal components  $M_2$  (1.93 cpd) and  $S_2$  (2.0 cpd) (Merritt, 2004; McMillan et al., 2019) were extracted from all time series and locations following the methods outlined in Schweizer et al. (2021) and Rau et al. (2020):

- The measured well water levels, barometric measurements and computed Earth tidal strain were de-trended using a moving linear regression filter with a 3-day window (Fig. 1d and the results are shown in Fig. A2.
- Amplitudes and phases of ten tidal harmonic constituents were jointly estimated using *Harmonic Least Squares* (HALS) (Fig. 1e (Schweizer et al., 2021). HALS was applied to the entire duration of the time series.
- From HALS, amplitudes and phases of the M2 and S2 tidal component were obtained for the Earth tide strains (Fig. 3a), barometric pressure (Fig. 3b) and hydraulic heads (Fig. 3c,d). tidal components were obtained for Earth tidal strains and hydraulic head. As the tidal component S2 encompasses both Earth tidal forces and atmospheric loading effects, the amplitudes and phases of the S2 tidal component were determined for Earth tide strains, hydraulic head, and barometric pressure.
- Complete disentanglement of the groundwater response to Earth and atmospheric tide influences was done for  $S_2$  following the method established by Rau et al. (2020) (Fig. 1f). This allows to separate the effects of Earth tides and atmospheric tides for the  $S_2$  constituent.

**RC:** What is the black line in Figure 5? It should be reflected in the title of the figure.

**AR:** We agree If the dots are located precisely on the black line, it indicates a complete alignment between the estimated properties obtained from both methods, signifying a perfect match. We added the line to the legend of the plot.

RC: L296-297 need to add references.

**AR:** We agree and added the references

Suggested revision at line 296: [...] There are several potential causes for this, such as the presence of flow paths that create undrained conditions, leading to a reduction in the generated confined pore pressure and exposing the limitations of passive methods for this borehole (Bastias et al., 2022; Wang, 2017; Cheng, 2016).

**RC:** It needs to be clarified whether the values calculated in L334-336 are the best fitting parameters.

**AR:** We agree and revised as follows:**

Suggested revision at line 334: [...] After constraining the solution space based on feasible values derived from the lithology information obtained from the well logs, 1,000 randomly generated values were employed as initial conditions for the least-squares algorithm. These values were generated according to a log-normal distribution. The purpose was to obtain an array of parameters that would converge to the best fit. The estimated values of aquifer hydraulic conductivity with Earth tidal analysis were  $1.1 \cdot 10^{-5} ms^{-1}$  and  $1.1 \cdot 10^{-4} ms^{-1}$  for borehole B1 and B2, respectively. Meanwhile, with the mean stress solution, the estimated values of aquifer hydraulic conductivity were  $1.6 \cdot 10^{-5} ms^{-1}$  and  $1.0 \cdot 10^{-4} ms^{-1}$  for borehole B1 and B2, respectively.

**RC:** The conclusion should include the proposal of future research.

**AR:** We agree and revised as follows:

Suggested revision at line 357: Analytical solutions assume simplified systems that often fail to comply with complex geologic formations. This discrepancy can result in significant errors when the assumptions of analytical solutions are violated. To assess the impact of such assumptions, two approaches can be considered. Firstly, numerical models can be employed to elucidate potential discrepancies between analytical and numerical solutions when the fundamental assumptions underlying analytical solutions are violated. Secondly, more intricate analytical or semi-analytical solutions can be developed that incorporate the mechanical effects of undrained conditions and/or consolidated systems.

**References**

- J. Bastias, G. C. Rau, and P. Blum. Groundwater responses to earth tides: Evaluation of analytical solutions using numerical simulation. *Journal of Geophysical Research: Solid Earth*, 127(10), sep 2022. doi: 10.1029/2022JB024771.
- A. H.-D. Cheng. Poroelasticity, volume 27. Springer, 2016.
- B. M. Das and B. Das. Advanced soil mechanics, volume 270. Taylor & Francis New York, 2008.
- B. G. Look. *Handbook of geotechnical investigation and design tables*. Taylor & Francis, 2007.
- T. C. McMillan, G. C. Rau, W. A. Timms, and M. S. Andersen. Utilizing the impact of earth and atmospheric tides on groundwater systems: A review reveals the future potential. *Reviews of Geophysics*, 57(2):281–315, 2019.
- M. L. Merritt. Estimating hydraulic properties of the Floridan aquifer system by analysis of earth-tide, ocean-tide, and barometric effects, Collier and Hendry Counties, Florida. Number 3. US Department of the Interior, US Geological Survey, 2004.
- G. Rau. Pygtide: A python module and wrapper for eterna predict to compute synthetic model tides on earth, zenodo [code], 2018.
- G. C. Rau, M. O. Cuthbert, R. I. Acworth, and P. Blum. Technical note: Disentangling the groundwater response to Earth and atmospheric tides to improve subsurface characterisation. *Hydrology and Earth System Sciences*, 24(12):6033–6046, 12 2020. ISSN 16077938. doi: 10.5194/hess-24-6033-2020. URL https://hess.copernicus.org/articles/ 24/6033/2020/.
- E. A. Roeloffs, S. S. Burford, F. S. Riley, and A. W. Records. Hydrologic effects on water level changes associated with episodic fault creep near Parkfield, California. *Journal of Geophysical Research*, 94(B9):12387, 1989. ISSN 01480227. doi: 10.1029/jb094ib09p12387. URL http://doi.wiley.com/10.1029/JB094iB09p12387.
- D. Schweizer, V. Ried, G. C. Rau, J. E. Tuck, and P. Stoica. Comparing Methods and Defining Practical Requirements for Extracting Harmonic Tidal Components from Groundwater Level Measurements. *Mathematical Geosciences*, 2 2021. ISSN 1874-8961. doi: 10.1007/s11004-020-09915-9. URL http://link.springer.com/10. 1007/s11004-020-09915-9.
- Z. Tickell. Water resources of the wildman river area. northern territory government, 2017. URL https://territorystories.nt.gov.au/10070/428586/0/53.

- R. Valois, G. C. Rau, J.-M. Vouillamoz, and B. Derode. Estimating hydraulic properties of the shallow subsurface using the groundwater response to earth and atmospheric tides: a comparison with pumping tests. *Water Resources Research*, 58(5):e2021WR031666, 2022.
- A. Verruijt. Theory and problems of poroelasticity. Delft University of Technology, 71, 2013.
- Wang. *Theory of linear poroelasticity with applications to geomechanics and hydrogeology*. Princeton University Press, 2017.
- H.-G. Wenzel. The nanoGal software: Earth tide data processing package: Eterna 3.3. *Bulletin d'Informations des Marées Terrestres*, 124:9425–9439, 1996.

---

## Author Comment (AC2)

**Response to Reviewer Comments 2**

We thank the reviewer for the comments. Note that we use the abbreviation **RC** for instance refer to reviewer comments 2, **EC** for editor and **AR** for our authors' response in the following. To highlight the nature of our reply, we use two colours; with green for agreement and yellow for misunderstanding. We explain all our reasoning and changes made to the manuscript. Removed text is shown in red, e.g., this text has been removed. New text is shown in blue, e.g., this text has been added.

**RC:** Overall, the manuscript is well written. The authors porposaed "mean stress solution" by introducing Biot's consolidation theory to estimate the groundwater fluctuation due to tidal loading.

**AR:** Thanks for the positive feedback, time spend and all the great comments that intend to improve our work.

**RC:** Line 110, it is state "Biot's consolidation theory assume  $\xi = 0$  when undrained condition. Why? please add reference.

**AR:** We agree: By definition,  $\xi$  represents the amount of fluid content in a system over a period of time (i.e.  $\frac{dm}{dt}$ , where *m* is mass of fluid in the system). Thus, if the system is undrained, which occurs when no fluid is entering or leaving a system,  $\xi = 0$ . We added references to this statement.

Suggested revision at line 110: Biot's consolidation theory assumes  $\xi = 0$  when undrained conditions apply within the porous medium (Cheng, 2016; Verruijt, 2013; Wang, 2017)

**RC:** Line 114: The assumption "exp(iwt)" implies only the first-order approximation is adopted. This requires justification.

**AR:** We agree: If the z-axis is vertical and the only body force is the atmospheric loading, the mechanical equilibrium equation reduces to,

$$\frac{\partial \sigma_{zz}}{\partial z} = -F_z \tag{1}$$

then  $\sigma_{zz}$  is independent of *z*, but it can be a function of time. Then the Mean Stress Equation in Cartesian coordinates can be written as,

$$\frac{\alpha}{KB} \left[ \frac{B}{3} \frac{d\sigma_{kk}}{dt} + \frac{\partial p}{\partial t} \right] - \frac{k}{\mu} \nabla^2 p = Q.$$
(2)

The quasi-static assumption of instantaneous mechanical equilibrium requires that the loading period be long relative to the times for elastic wave propagation (which is the case for tides). Under this pseudo steady-state approximation, time and z are independent and the solution is first order,

$$p(z,t) = \tilde{p}(z)exp(i\omega t)$$
(3)

where  $\tilde{p}(z)$  is the (complex) amplitude, which depends only on z.

Suggested revision at line 114: We solved Eq. 1 for steady state conditions to obtain the periodic water level in an open borehole. This assumes that the instantaneous mechanical equilibrium requires the loading period to be long relative to the times for elastic wave propagation. Furthermore, the initial transient when the surface loading starts is neglected. Under this pseudo steady-state approximation, i.e., time and depth are independent, the solution is first order and has the form  $h_w^{AT} = h_{w,o}^{AT} e^{i\omega t}$ due to atmospheric loading, where  $\omega$  is the angular frequency of the tide signal and superscript *AT* stands for atmospheric tides, for example  $S_1$  at 1 cycle per day (CPD) or the atmospheric response to  $S_2$  at 2 cpd (Merritt, 2004; McMillan et al., 2019). **RC:** The assumption of "Mean stress" needs justification.

**AR:** We agree: The poroelastic constitutive relationship is

$$\xi = \frac{\alpha}{K} \frac{\sigma_{kk}}{3} + \frac{\alpha}{KB} p, \tag{4}$$

wWhere  $\alpha$  represents the Biot conefficient, *K* the bulk modulus, *B* the Skeptom coefficient and  $\sigma_{kk}$  the mean stress. If a change in pore pressure is only cause by a vertical loading and only vertical deformation is allowed, then

$$\frac{\sigma_{kk}}{3}_{\varepsilon_{xx}=\varepsilon_{yy}=\xi=0} = \frac{1}{3} \frac{1+v_u}{1-v_u} \sigma_{zz}.$$
(5)

Now, the fluid continuity equation is

$$\frac{\partial \xi}{\partial t} - \frac{k}{\mu} \nabla^2 p = Q. \tag{6}$$

Substituting for  $\xi$  from the constitutive equation Eq. 4 gives

$$\frac{\alpha}{KB} \left[ \frac{B}{3} \frac{\partial \sigma_{kk}}{\partial t} + \frac{\partial p}{\partial t} \right] - \frac{k}{\mu} \nabla^2 p = Q, \tag{7}$$

which is presented as Mean Stress Equation.

•

Suggested revision at line 102: [...] For example, when atmospheric pressure rises and, the formation undergoes compressive stress resulting in an increased in the confined pore pressure (Domenico and Schwartz, 1997). If a change in pore pressure is only caused by a vertical loading and only vertical deformation is allowed, then the mean stress can be computed as

$$\frac{\sigma}{3} \sum_{\varepsilon_{xx}=\varepsilon_{yy}=\xi=0} = \frac{1}{3} \frac{1+v_u}{1-v_u} \sigma_{zz}.$$
(8)

**RC:** In this note, only tow borehole data are used. As shown in Figure 5(c), two data point is obviously insufficient. The authors may need to add more data points.

**AR:** We partially agree, it is indeed acknowledged that relying on just two data points is inadequate, and a more extensive investigation is necessary to thoroughly evaluate the reliability and performance of the analytical solution using field data. The primary aim of this paper is to develop a new analytical solution. In addition, we have applied this to field data and validated its results by comparison with established solutions. Further, we present a comprehensive workflow for signal treatment of field data, with a particular emphasis on the utilisation of the analytical solution, rather than focusing solely on the field data itself.

**References**

- A. H.-D. Cheng. Poroelasticity, volume 27. Springer, 2016.
- P. A. Domenico and F. W. Schwartz. *Physical and chemical hydrogeology*. John wiley & sons, 1997.
- T. C. McMillan, G. C. Rau, W. A. Timms, and M. S. Andersen. Utilizing the impact of earth and atmospheric tides on groundwater systems: A review reveals the future potential. *Reviews of Geophysics*, 57(2):281–315, 2019.
- M. L. Merritt. *Estimating hydraulic properties of the Floridan aquifer system by analysis of earth-tide, ocean-tide, and barometric effects, Collier and Hendry Counties, Florida.* Number 3. US Department of the Interior, US Geological Survey, 2004.
- A. Verruijt. Theory and problems of poroelasticity. Delft University of Technology, 71, 2013.
- Wang. *Theory of linear poroelasticity with applications to geomechanics and hydrogeology*. Princeton University Press, 2017.